



# Limited exchange between the deep Pacific and Atlantic oceans during the warm mid-Pliocene and MIS M2 "glaciation"

Anna Hauge Braaten[1], Kim A. Jakob[2], Sze Ling Ho[1,3], Oliver Friedrich[2], Eirik Vinje Galaasen[1], Stijn De Schepper[4], Paul A. Wilson[5] and Anna Nele Meckler[1,]

[1]Department of Earth Science and Bjerknes Centre for Climate Research, University of Bergen, Bergen, 5007, Norway
[2]Institute of Earth Sciences, Heidelberg University, Heidelberg, 69120, Germany
[3]Institute of Oceanography, National Taiwan University, Taipei, 10617, Taiwan
[4]NORCE Norwegian Research Centre and Bjerknes Centre for Climate Research, Bergen, 5007, Norway
[5]University of Southampton, Waterfront Campus, National Oceanography Centre, Southampton, SO14 3ZH, UK

*Correspondence to*: Anna H. Braaten (anna.braaten@uib.no)

**Abstract**

The mid-Pliocene (3.3-3.0 Ma) is the most recent period in Earth's history of sustained, global warmth analogous to predicted near-future climates. Despite considerable efforts to characterize and understand the climate dynamics of the mid-Pliocene, the deep ocean and its response to this warming remains poorly understood. Here we present new mid-Pliocene Mg/Ca and $\Delta_{47}$ ("clumped isotope") temperatures from the deep Pacific and North Atlantic oceans. These records cover the transition from Marine Isotope Stage (MIS) M2 — considered the most pronounced "glacial" stage of the mid-Pliocene — to the warm KM5 interglacial. We find that a large (>4°C) temperature gradient existed between these two basins throughout that interval, with the deep North Atlantic considerably warmer and likely saltier than at present. We interpret our results to indicate that the deep Pacific and North Atlantic oceans were bathed by water masses with very different physical properties during the mid-Pliocene, and that only limited deep oceanic exchange occurred between the two basins. Our results point to a fundamentally different mode of ocean circulation or mixing compared to the present, where heat and salt is distributed from the North Atlantic into the Pacific. The amplitude of cooling observed at both sites during MIS M2 suggests that changes in benthic $\delta^{18}$O associated with this cold stage were mostly driven by temperature change in the deep ocean rather than ice volume.

## 1 Introduction

Our ability to predict future climate change in response to anthropogenic $CO_2$ emissions partially rests on our understanding of how the climate system has operated under similar conditions in the past. A period that has received considerable attention as a useful point of comparison for near-future climates is the mid-Pliocene ( 3.3–3.0 million years ago, Ma), when atmospheric $CO_2$ was comparable to present values (~350–450 ppm, de la Vega et al., 2020 and references therein) and many important tectonic and geographical boundary conditions were similar to today.



Considerable efforts have been made to characterize and understand the climate dynamics of the mid-Pliocene. Globally averaged surface temperatures were elevated by ~2–3 °C relative to the pre-industrial (McClymont et al., 2020; Haywood et al., 2020), with much of that warming concentrated in the high latitudes. Meanwhile, maximum temperatures in the tropical warm pools appear to have been comparable to today (~29 °C) (Wara et al., 2005; Meinicke et al., 2021), resulting in significantly reduced meridional temperature gradients. Sea-level rise of the magnitude proposed for the mid-Pliocene (~20 meters above present) (Dwyer and Chandler, 2009; Rohling et al., 2014; Grant et al., 2019; Dumitru et al., 2019) would require reduced ice sheet extent in both hemispheres. In equilibrium with $CO_2$ concentrations comparable to the early 21[th] Century, it appears that the climate of the mid-Pliocene was warm enough to prevent growth of significant continental ice in the Northern Hemisphere (Dumitru et al., 2019).

Earth system models have largely been unable to simulate the full magnitude of Arctic amplification and reduced meridional temperature gradients implied by proxy reconstructions (Dowsett et al., 2012; Salzmann et al., 2013; de Nooijer et al., 2020), suggesting that key feedback mechanisms for the mid-Pliocene may be underestimated or missing in these models. Increased poleward heat transport, decreased ice-albedo and feedbacks related to cloud-cover are some of the mechanisms proposed to have contributed to Pliocene warmth (Fedorov et al., 2006), but a full understanding of the processes that caused and maintained these conditions is still missing.

While climatic conditions of the mid-Pliocene were typically warm, this interval also contains distinct climate variability. Most notably, it includes a short-lived but pronounced "glacial" event during Marine Isotope Stage (MIS) M2 (3.312–3.264 Ma). MIS M2 is the largest positive benthic oxygen isotope excursion (~0.6 ‰) in the Pliocene prior to the intensification of Northern Hemisphere glaciation (iNHG) (Lisiecki and Raymo, 2005), and has been suggested to represent an early, "failed attempt" at establishing a pattern of Northern Hemisphere glacial-interglacial cycles (Haug and Tiedemann, 1998). Estimates of sea-level fall associated with this event vary greatly, ranging from ~10 to 65 meters below present level (Dwyer and Chandler, 2009; Naish and Wilson, 2009; Miller et al., 2012). To what extent the oxygen isotope event reflects ice growth on land versus cooling of the deep ocean, however, remains a topic of debate.

The deep ocean plays an integral role in modulating climate on long and short timescales, acting as a major reservoir for heat and $CO_2$ that responds to and affects surface conditions. Although the mid-Pliocene deep ocean remains poorly characterized, with available temperature records sparse and occasionally contradictory, existing reconstructions suggest the deep ocean could have operated differently than today. In the North Atlantic, available data suggest average bottom water temperatures (BWTs) were either similar to today (Dwyer et al., 1995; Cronin et al., 2005; Dwyer and Chandler, 2009) or significantly warmer (Bartoli et al., 2005; Sosdian and Rosenthal, 2009). The deep North Pacific, however, may have been — on average — colder than today in the mid-Pliocene (Woodard et al., 2014). Based on comparison of this North Pacific record (Site 1208,



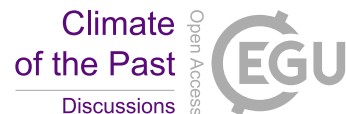

3346 meters water depth) with North Atlantic BWTs from Site 607 (3426 meters water depth, Sosdian and Rosenthal, 2009),

Woodard et al. (2014) conclude that a large bottom water temperature gradient of up to 4 °C existed between the two basins

in the Pliocene prior to the iNHG at ~2.7 Ma. In comparison, the modern deep ocean is relatively isothermal across the various

basins, with only a ~1°C difference between the deep (>3000 meters) Atlantic, Pacific and Indian Oceans (Locarnini et al.,

2013) (Fig. 1). The existence of a strong temperature gradient in the mid-Pliocene would have important implications for

characterising ocean circulation and global climate, determining for example where heat resided in the climate system and the

route and efficiency of heat transport. A growing number of studies have suggested that the North Pacific Ocean was a site of

deep convection and deep water formation (North Pacific Deep Water, NPDW) (Burls et al., 2017; Shankle et al., 2021; Ford

et al., 2022) in the warm Pliocene. If true, this would leave open the possibility that the records of Woodard et al. (2014) from

Shatsky Rise, which suggest colder-than-present temperatures in the mid-Pliocene, are recording a local North Pacific signal

rather than representing the deep Pacific as a whole.

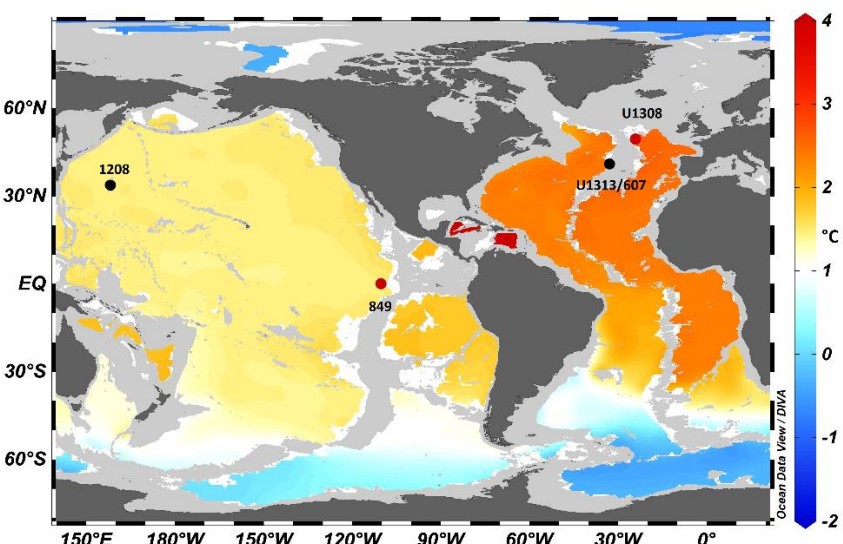

**Figure 1. Modern water temperatures at 3800 meters water depth with locations of sites used (in red) and referenced (in black). Temperature data from the World Ocean Atlas 2013 (Locarnini et al., 2013). Map generated in Ocean Data View (ODV 5.6.3,**
**Schlitzer, 2023)**

Available BWT records from the mid-Pliocene have exclusively been based on Mg/Ca thermometry. The Mg/Ca

paleothermometer is influenced by a number of non-thermal effects that complicate the application of this proxy, which could

explain some of the discrepancies observed between various mid-Pliocene BWT records. Mg/Ca ratios of epifaunal benthic

foraminifera can be affected by changes in carbonate ion saturation state (Elderfield et al., 2006; Lear et al., 2010), and it has

been suggested that the North Atlantic Mg/Ca-based temperature record by Sosdian and Rosenthal (2009) was compromised

by such effects (Yu and Broecker, 2010). Additionally, strong vital effects have been documented in foraminifera — making



species-specific calibrations necessary (i.e. Lear et al., 2002), and some of the Mg/Ca-based estimates are from ostracods, possibly explaining discrepancies from foraminiferal records. Furthermore, seawater Mg/Ca (Mg/Ca$_{sw}$), while globally

uniform at any given time, changes on geological timescales (e.g. Wilkinson and Algeo, 1989; Dickson, 2004; Coggon et al., 2010). Adjustments for this change on long time scales have been applied since the first applications of the Mg/Ca BWT proxy (Lear et al., 2000), but estimates of the magnitude of change — and thus its effect on reconstructed temperatures — varies between studies (Evans et al., 2016; Tierney et al., 2019).

A novel approach for estimating BWTs in the Pliocene is carbonate clumped isotope ($\Delta_{47}$) thermometry of benthic foraminifera. This paleothermometer is based on the thermodynamic preference for rare, heavy isotopes to bond within the same carbonate ion with decreasing temperatures, and involves measuring the excess abundance of $^{13}$C-$^{18}$O bonds relative to their stochastic abundance (Ghosh et al., 2006; Schauble et al., 2006). This proxy produces temperature estimates that are independent of the chemical composition of the parent water (e.g. Eiler, 2011), making it a particularly powerful tool for

reconstructing ocean temperatures in environments where the composition of seawater cannot be well constrained. Furthermore, no detectable species-specific vital effects have been found for benthic foraminifera (Tripati et al., 2010; Peral et al., 2018; Piasecki et al., 2019). While $\Delta_{47}$ is a highly useful proxy for reconstructing absolute temperatures, the large quantities of carbonate sample material required for reliable temperatures coupled with the large analytical uncertainties and time-intensive nature of the measurements, makes it less suited for high-resolution work.


To address the discrepancies between currently available deep-sea temperature records for the mid-Pliocene and assess the existence of a large temperature gradient between the Pacific and Atlantic basins we present new paired benthic foraminiferal Mg/Ca and $\Delta_{47}$ temperatures from Ocean Drilling Program (ODP) Site 849 in the deep Equatorial East Pacific and Integrated Ocean Drilling Program (IODP) Site U1308 in the deep North Atlantic. The records cover the interval from Marine Isotope

Stage (MIS) M2 — considered the most pronounced "glacial" stage of the mid-Pliocene — to the warm interglacial MIS KM5c, which has a near identical orbital configuration to the present and thus possibly offers the best analogue for our current climate (i.e. Haywood et al., 2013). This allows us to not only compare mean deep-sea temperatures in the Pacific and Atlantic basins, but also to investigate the variability over key warming and cooling stages. Because Mg/Ca records at both sites were generated on the same shallow-infaunal benthic foraminiferal species (*Oridorsalis umbonatus*), and samples and data were

treated identically, the records are more directly comparable than those of previous studies. $\Delta_{47}$ temperatures measured on the same samples provide an independent check on absolute Mg/Ca temperatures, bypassing uncertainties related to seawater chemistry or species effects. In combination, the two proxies provide robust constraints on bottom-water temperatures in the deep Pacific and North Atlantic for the mid-Pliocene and allows us to examine if changes in benthic $\delta^{18}$O associated with MIS M2 are caused by temperature or ice volume.



## 2 Materials and Methods

### 2.1 Materials, site locations and sample processing

#### 2.1.1 ODP Site 849

ODP Site 849 is situated on the western flank of the East Pacific Rise (Fig. 1) (0°11'N, 110°31'W) at a water depth of 3851 meters (Mayer et al., 1992). It is today bathed by Pacific Deep Water (PDW), a water mass largely consisting of Antarctic Bottom Water (AABW), with smaller contributions from Antarctic Intermediate Water (AAIW) and recirculated North Atlantic Deep Water (NADW) (Mix et al., 1995; Kwiek and Ravelo, 1999). This study site was chosen as it is considered to approximate mean deep Pacific conditions, and is suggested to have been permanently bathed by southern component waters throughout the Plio-Pleistocene (Mix et al., 1995). Although the present day water depth at Site 849 sits below the modern eastern equatorial lysocline at ~3400 (Berger et al., 1982) the site has remained above the carbonate compensation depth for the past 34 Ma (Pälike et al., 2012). Previous studies have concluded that this site exhibits good foraminiferal test preservation, excellent age control and high sedimentation rates across our study interval (Jakob et al., 2021a, Jakob et al., 2021b).

We investigated samples along the primary shipboard splice, from cores 849D-8H-5-6 cm to 849D-8H-6-96 cm and 849C-9H-1-46 cm to 849C-9H-3-18 cm (87.67–92.77 meters composite depth, mcd), corresponding to 3.160–3.334 Ma on the updated age model of Jakob et al. (2021b). A total of 248 (20 cm³) samples were collected at 2-cm intervals, and the material was dried, weighed and washed over a 63 μm sieve. Tests of the benthic foraminifer *O. umbonatus* were picked from the >150 μm dried sediment fraction for Mg/Ca analysis. In addition, remaining benthic foraminifera were picked from selected samples (n=44, >150 μm fraction) for clumped isotope analysis.

#### 2.1.2 IODP Site U1308

IODP Site U1308 constitutes a reoccupation of Deep Sea Drilling Project (DSDP) Site 609, a benchmark location for late Pleistocene North Atlantic climate records. Site U1308 is situated on the eastern flank of the Mid-Atlantic Ridge (Fig. 1) (49°52'N, 24°14'W) at a water depth of 3871 meters (Expedition 303 Scientists, 2006). Today, this site is bathed by Lower North Atlantic Deep Water (LNADW) and Lower Deep Water (LDW), with LDW consisting of a mixture of 70–80 % NADW and 20–30% AABW (Arhan et al., 2003). Site U1308 was chosen as it is situated at the same depth as our Pacific site and exhibits good foraminiferal test preservation for our target interval (Fig. S4).

We investigated samples from cores U1308C-25H-5-80 cm to U1308C-26H-6-71.5 cm (253.81–264.71 mcd). Very little work has been done on this portion of the sequence because it falls below the primary shipboard splice (0–248 mcd, ~0–3.1 Ma). However, an age model and benthic foraminiferal (*Cibicidoides wuellerstorfi* and *Uvigerina peregrina*) stable isotope data are available for a short interval from 258.95–264.71 mcd, corresponding to MIS M2 (De Schepper et al., 2013). To extend this record, 53 (20 cm³) samples were collected at 10 cm spacing from 253.81–259.01 mcd. The new samples were dried, weighed and washed over a 63 μm sieve. The >150 μm sediment fraction of each of these samples was picked for *O. umbonatus* and *C.*





*wuellerstorfi*/*Cibicidoides mundulus* for Mg/Ca and stable isotope analysis, respectively. The interval from 258.95–264.71 mcd was sampled by De Schepper et al. (2013) at a higher resolution, on average every 5 cm (n=113). These samples were picked for *O. umbonatus* for Mg/Ca analysis. Furthermore, remaining benthic foraminifera were picked from 34 samples (>150

μm) across the full study interval for clumped isotope analysis.

## 2.2 Methods

### 2.2.1 Stable isotope analysis and age model for Site U1308

Stable isotope ($\delta^{18}O$ and $\delta^{13}C$) analyses were performed on both *C. wuellerstorfi* and *C. mundulus*, as neither species was present continuously throughout the entire study interval. Prior to analysis, foraminiferal tests were ultrasonicated in methanol

for 15 seconds to remove fine-grained particles. Analyses were carried out at the Facility for Advanced Isotopic Research and Monitoring of Weather, Climate and Biogeochemical Cycling (FARLAB), Department of Earth Science, University of Bergen, using a Finnigan MAT 253 mass spectrometer coupled to a Kiel IV carbonate device. Measurements were performed on 1–3 tests and replicated two or more times when abundances allowed (~76% of the 47 samples measured). Six samples did not contain enough material for analysis. The long-term reproducibility (1σ) of the in-house working standard (CM12, Carrara

marble) during the analysis window was 0.03‰ and 0.06‰ for $\delta^{13}C$ and $\delta^{18}O$, respectively. Results are reported relative to Vienna Pee Dee Belemnite (VPDB), calibrated using National Bureau of Standards (NBS)-19 and crosschecked with NBS-18. We detect no systematic variations in oxygen isotopic offsets between the two species and present the data as an averaged composite record. An apparent inter-lab offset between previously the published benthic foraminiferal $\delta^{18}O$ ($\delta^{18}O_b$) of De Schepper et al. (2013) and our new record was observed. We adjust for this offset by adding 0.29‰ to the values reported in

De Schepper et al. (2013) (see supplementary materials and Fig. S1 for details).

**Table 1. Tie-points used for the updated age model of IODP U1308**

| Depth (mcd) | Age (Ma) |
|---|---|
| 254.66 | 3.207 |
| 256.77 | 3.240 |
| 261.11 | 3.284 |
| 262.35 | 3.302 |
| 263.25 | 3.320 |
| 263.38 | 3.327 |
| 264.70 | 3.340 |



For Site U1308, a continuation of the existing age model by De Schepper et al. (2013) was established for the interval from
253.81–258.95 mcd by tuning our new $\delta^{18}O_b$ record to the LR04 benthic foraminiferal oxygen isotope stack (Lisiecki and
Raymo, 2005) using the software program QAnalySeries 1.5.0.(Kotov and Pälike, 2018). Minor adjustments were also made
to the existing age model between 258.95–264.71 mcd. The updated tie-points used for the age model are presented in Table
1.

**2.2.2 Mg/Ca temperatures**

The benthic foraminiferal species *O. umbonatus* was used for Mg/Ca analysis at both sites. This species was selected because
i) it is well buffered against the influence of changes in deep ocean carbonate ion/pH due to its shallow-infaunal habitat
(Rathmann and Kuhnert, 2008; Lear et al., 2015), ii) it contains large chambers that can be thoroughly cleaned for Mg/Ca
analysis, and iii) it exhibits low sensitivity to temporal variations in the Mg/Ca of seawater (Lear et al., 2015). Furthermore,
core top measurements of this species from Site 849 (Jakob et al., 2021a) and North Atlantic Site U1313 (Jakob et al., 2020
suppl. information), have produced Mg/Ca temperatures that are indistinguishable from modern BWTs at the respective sites.
These core top studies treated materials identically to our mid-Pliocene samples.

Foraminiferal tests were cracked, homogenized and cleaned following the cleaning protocol of Barker et al. (2003) with the
reductive cleaning step omitted to avoid decreasing their Mg/Ca ratios (Barker et al., 2003; Rosenthal et al., 2004). Samples
were measured using an Agilent Inductively Coupled Plasma-Optical Emission Spectrometer (ICP-OES) 720 at the Institute
of Earth Sciences, Heidelberg University. Reported Mg/Ca values have been normalized relative to the ECRM 751-1 standard
(Greaves et al., 2008). Fe/Ca and Mn/Ca ratios were screened to identify potential contamination from clays or coatings that
were not removed in the cleaning process (see suppl. information for details). Fe/Ca and Mn/Ca ratios were not normalized as
the Fe and Mn concentrations of the ECRM standard typically were below the detection limit of the ICP-OES. The ECRM
standard was measured at least every 20[th] sample to monitor instrumental precision. Based on these replicate measurements,
the standard deviation for Mg/Ca is ± 0.02 mmol/mol and ± 0.03 mmol/mol for Site 849 and Site U1308 samples, respectively.

Bottom water temperatures were calculated using the *O. umbonatus*-specific calibration by Lear et al. (2002):

$$T = \left[ \ln \frac{Mg/Ca}{1.008} \right] \times \left[ \frac{1}{0.114} \right] \qquad\qquad (1)$$

This calibration is based on multiple, globally distributed core tops, making it applicable to both study sites. Furthermore, the
calibration temperature range of 0.8–9.9 °C covers the temperature range we expect to find in the mid-Pliocene deep ocean.
Because this calibration was based on data from materials cleaned following a procedure that includes a reductive step,
measured Mg/Ca was adjusted downwards by 10% to account for this difference (Barker et al., 2003; Ford et al., 2016). Values





were also adjusted to account for past variation in Mg/Ca$_{sw}$ following Lear et al. (2002) using estimates of past Mg/Ca$_{sw}$ from Evans et al. (2016). Error propagation including both analytical and calibration uncertainty produces an error associated with our Mg/Ca-based BWT record of ± 0.9 °C and ± 1.5 °C for Site 849 and U1308 samples, respectively.


### 2.2.3 Δ$_{47}$ temperatures

Before clumped isotope (Δ$_{47}$) analysis, foraminiferal tests were carefully cracked open to expose the inside of individual chambers. The broken open tests were ultrasonicated for 10 s in DI water, then 10 s in methanol, followed by another two ultrasonication steps with DI water. Samples were rinsed with DI water after each ultrasonication. Cleaned samples were oven-

dried at 50°C until any remaining water was fully evaporated. Scanning electron microscope (SEM) images (Fig. S4 and S5) were taken from a random selection of cleaned samples to verify i) that cleaning had fully removed potential contaminants and ii) that tests were well-preserved.

Clumped isotope measurements were performed using two different Thermo Scientific MAT 253Plus mass spectrometers

coupled to Kiel IV carbonate devices, located at FARLAB, Department of Earth Science, University of Bergen.
The analytical approach is described in detail by Meckler et al. (2014), Piasecki et al. (2019) and Meinicke et al. (2020). To remove potential organic contaminants, the Kiel devices were equipped with PoraPak traps which were held at -20°C during runs. The traps were baked at 150°C for at least one hour between runs for cleaning. A mix of samples and carbonate standards in a roughly 1:1 ratio was measured in each analytical run. Micro-volume aliquots (70-100 µg) were individually reacted with

phosphoric acid (at 70°C) in the Kiel device, and the resulting gas was subsequently measured using the long-integration dual-inlet method (LIDI, Hu et al. 2014) for a total of 400 s. Raw data were corrected for pressure baseline effects based on five daily peak scans (5–25 V, Meckler et al. 2014). Using carbonate standards ETH 1–3, the data were further corrected for scale compression and transferred to the I-CDES scale (Bernasconi et al., 2021). Standard data from the same and adjacent days were used for data correction, using a moving window approach. All data correction was done using the Easotope software

package (John and Bowen, 2016).

Due to the large analytical uncertainty associated with individual Δ$_{47}$ measurements, extensive replication (preferably a minimum of 25–30 measurements) is required to produce reliable temperatures. A large number of benthic foraminiferal species were used to obtain sufficient amounts of sample material (2–4 mg) needed per Δ$_{47}$ temperature. Whenever possible,

measurements were performed on aliquots of species- or genus-specific materials. Replicate measurements (n=23–40) from 2–9 adjacent samples were combined to produce each data point. Temperatures were calculated from these averaged Δ$_{47}$ values using the combined foraminifera-based calibration by Meinicke et al. (2020), updated to the I-CDES scale by Meinicke et al. (2021):





$$T = \sqrt{\frac{0.0397 \times 10^6}{\Delta^{47} - 0.1518} - 273.15}$$  (2)

With temperatures given in °C.

The combined analytical and calibration uncertainty was calculated using a Monte Carlo approach and is expressed as 95%
confidence intervals on the average temperatures.

### 2.2.4 Seawater $\delta^{18}O$

Seawater $\delta^{18}O$ ($\delta^{18}O_{sw}$) was calculated from $\delta^{18}O_b$ in combination with the BWT from Mg/Ca using the *Cibicidoides* and
*Planulina* compilation (Eq. 9) from Marchitto et al. (2014):

$$\delta^{18}O_{sw} = [0.245 \times T] - [0.0011 \times T^2] + \delta^{18}O_b - 3.31$$  (3)

Due to the good agreement between the absolute temperatures suggested by the two proxies at both sites (see results), we
calculate $\delta^{18}O_{sw}$ from BWT estimates from Mg/Ca, not $\Delta_{47}$, as i) the Mg/Ca records are of significantly higher resolution and
ii) the uncertainties associated with individual Mg/Ca measurements (±0.8°C and ±1.5°C for 849 and U1308 samples
respectively) are smaller than those for the $\Delta_{47}$-based estimates (average of ±2.7°C at both sites). For $\delta^{18}O_{sw}$ calculation we
use the published $\delta^{18}O_b$ from Jakob et al. (2021b, generated on *O. umbonatus*) for Site 849 and for Site U1308 the combined
record from this study (*C. wuellerstorfi* and *C. mundulus)* and De Schepper et al. (2013) (*C. wuellerstorfi* and *Uvigerina
peregrina*), with a +0.29‰ correction of values from the latter (see suppl. Information). To account for species-specific offsets,
we normalize measured $\delta^{18}O_b$ of *O. umbonatus* and *Uvigerina spp*. to *Cibicidoides* by subtracting 0.64‰ (Shackleton et al.,
1984), allowing us to apply Eq. 3 to all samples.

### 3 Results

#### 3.1 Pacific Site 849

A total of 233 Mg/Ca measurements were carried out for the interval 3.334–3.160 Ma, yielding a mean temporal resolution of
800 yr. The five-point moving average of Mg/Ca-based BWTs ranges between -0.3°C and 4.6°C (Fig. 2b) while individual
values vary from -1.3°C to 6.9°C. The equation used to calculate these temperatures has a calibration range of 0.8–9.9 °C (Lear
et al., 2002). In total, 42 (~5%) of our measurements fall below this range (-1.3–0.8°C) and are therefore associated with
greater uncertainty than the remainder of the record. However, the same would also be true for all other currently available *O.
umbonatus*-specific Mg/Ca calibrations (see Jakob et al., 2021a and references therein for a review).








**Figure 2. Climate indicators and results from Pacific ODP Site 849. All horizontal stippled lines indicate modern values ($CO_2$ = 2022). (a) Benthic foraminiferal $\delta^{18}O$ from Site 849 (Jakob et al. 2021b, shown with 5pt running mean) and global benthic foraminiferal $\delta^{18}O$ stack (Lisiecki and Raymo, 2005). (b) 849 temperature records. Mg/Ca temperatures shown with 5pt running mean. Horizontal error bars on $\Delta_{47}$ temperatures indicate the age range of all individual samples used for each data point. Temperature uncertainties are expressed as solid (68% CI) and stippled (95% CI) bars. (c) Interpolated 1 ka resolution $\delta^{11}B$-derived $CO_2$ estimates from de la Vega et al. (2020). Confidence intervals expressed as dark (68%) and light (95%) shading. (d) Summer insolation forcing at 65°N and 65°S (Laskar et al., 2004). (e) calculated $\delta^{18}O_{sw}$, shown with a linear regression to highlight the overall trend (shading = 95% CI). Note arrows indicating the temporal offsets in the increase in $\delta^{18}O_b$ and decrease in temperature and $CO_2$.**

The average temperature over the study interval is ~2.5C±0.8°, 1°C warmer than present BWT at this site (1.6°C, Locarnini et al., 2013) but large changes in BWT are observed throughout the record. Peak temperatures (>4°C) are recorded during all the studied interglacial (as defined from $\delta^{18}O_b$) Marine Isotope Stages (MG1, M1, KM5 and KM3). We observe strong cooling (by ~4°C) associated with the large positive $\delta^{18}O_b$ excursion during MIS M2, although the cooling appears significantly delayed relative to the signal in $\delta^{18}O_b$. Cooling of similar amplitude is also recorded for KM6 and KM4, Marine Isotope Stages not appearing as pronounced glacial stages in benthic oxygen isotope records (Fig. 2a and b).

Ten clumped isotope temperatures were generated over the interval from 3.328–3.189 Ma (Fig. 2b), with each data point representing an average over 23–40 replicate measurements from 2–9 adjacent samples (time interval spanned indicated by horizontal bars in Fig. 2b). $\Delta_{47}$-based BWTs range from 0.7°C±1.9°C to 5.4°C±2.3°C (95% CI). The average $\Delta_{47}$-based BWT over the studied interval is 3.5°C±0.8°C.

Our calculated $\delta^{18}O_{sw}$ record (Fig. 2e) indicates an average value of ~0.0‰ (present bottom water $\delta^{18}O_{sw}$ at Site 849: -0.05‰, LeGrande and Schmidt, 2006) over the study interval. Although $\delta^{18}O_{sw}$ is variable, there is a clear trend of decreasing values over the record — values are higher before and during MI2 M2 than afterwards. Average $\delta^{18}O_{sw}$ from 3.334–3.264 Ma (MIS M2 and MG1) is ~+0.3‰. Between 3.263 and 3.160 Ma (MIS M1–KM3), the average $\delta^{18}O_{sw}$ is ~-0.1‰.

## 3.2 North Atlantic Site U1308

A total of 49 Mg/Ca measurements were performed for the interval ~3.334–3.196 Ma, yielding a mean temporal resolution of ~3 kyr. The three-point smoothed average of Mg/Ca-based bottom-water temperatures ranges from 5.3–9.3°C, with individual values between 4.5–11.7°C (Fig. 3b). A single measurement (11.7°C) falls outside the calibration temperature range of 0.8–9.9°C. The average BWT across the study interval is ~7.2±1.5°C, approximately 4.5°C warmer than the present BWT at this site (2.6°C, Locarnini et al., 2013). The coldest temperatures of the record are reached during MIS M2, but similar temperatures are also reconstructed for MIS KM6. However, the temporal resolution across the most intense $\delta^{18}O_b$ excursion of MIS M2 is low, and our record may underestimate the temperature change associated with this event.









**Figure 3. Climate indicators and results from North Atlantic IODP Site U1308. All horizontal stippled lines indicate modern values**
**(CO2 = 2022). (a) Benthic foraminiferal δ¹⁸O from Site U1308 (shown with 3pt running mean and global benthic foraminiferal δ¹⁸O**
310 **stack. (b) U1308 temperature records Mg/Ca temperatures shown with 3pt running mean. Horizontal error bars on Δ47 temperatures**
**indicate the age range of all individual samples used for each data point. Temperature uncertainties are expressed as solid (68% CI)**
**and stippled (95% CI) bars. (c) Interpolated 1 ka resolution δ¹¹B-derived CO2 estimates from de la Vega et al. (2020). Confidence**
**intervals expressed as dark (68%) and light (95%) shading. (d) fish debris Nd isotope data from North Atlantic IODP Site U1313**
**(Lang et al., 2016; Kirby et al., 2020). Estimates of SCW and NCW end-member εNd composition following Lang et al., (2016). (e)**
315 **calculated δ¹⁸Osw shown with a linear regression to highlight the overall trend (shading = 95% CI ).**

Eight clumped isotope temperatures were generated for the interval from 3.335–3.199 Ma (Fig. 3b), each calculated from an average of 27–34 replicate measurements from 2–7 neighboring samples. $\Delta_{47}$-based BWTs range from 5.6°C±2.7°C to 10.5°C±3.4°C (95% CI), with both the warmest and coldest data points falling within different stages of MIS M2. The average 320 $\Delta_{47}$-based BWT across the full study interval is 7.7°C±1.0°C (95% CI).

Our calculated record of $\delta^{18}O_{sw}$ (Fig. 3e) indicates an average of 1.1.‰ over the study interval, considerably higher than the present bottom water $\delta^{18}O_{sw}$ at Site U1308 of 0.25‰ (LeGrande and Schmidt, 2006). Similarly to Site 849, there is an apparent trend of decreasing values over the record — with somewhat higher $\delta^{18}O_{sw}$ before and during MIS M2 than afterwards. 325 However, the significantly lower temporal resolution of the Site U1308 $\delta^{18}O_{sw}$ record makes this finding more uncertain than is the case for the Pacific.

## 4 Discussion

### 4.1 Comparison between Mg/Ca and Δ47 temperatures

Comparison of our Mg/Ca- and $\Delta_{47}$-based BWT records reveal good proxy agreement at both our Pacific and North Atlantic 330 sites. At Pacific Site 849, all $\Delta_{47}$ temperatures are within error (95% CI) of the Mg/Ca-based record (Fig. 2b). The two proxies suggest similar average temperatures (Mg/Ca=~2.5°C±0.8°C, $\Delta_{47}$=3.5°C±0.8°C), 1–2°C warmer than modern BWT at this site. At North Atlantic Site U1308, both proxies record similar average temperatures (Mg/Ca=7.2°C±1.5°C, $\Delta_{47}$=7.7°C±1.0°C) approximately 4.5°C warmer than modern BWTs at this location (Fig. 3b). The $\Delta_{47}$-based temperature at 3.280 Ma (10.5°C±3.4°C) is considerably warmer than three Mg/Ca-derived temperatures generated on some of the same samples (6.2– 335 7.4°C). We find no obvious explanation for this discrepancy. None of the individual $\Delta_{47}$ replicates (n=32) are classified as outliers according to our criteria (4x SD), and the $\delta^{18}O_b$ values measured alongside the $\Delta_{47}$ are in excellent agreement with $\delta^{18}O_b$ measured separately (Fig. 4). It is possible that our Mg/Ca is underestimating BWT warming at the termination of M2, although we find this explanation to be unlikely. Extreme bottom-water warming at this time is not supported by either of the proxy records from the Pacific (Fig. 2b), or by the $\delta^{18}O_b$ records from either site (Fig. 2a and Fig. 3a). Furthermore, no available 340 SST records from the high-latitude North Atlantic indicate such warming (e.g. Lawrence et al., 2009; Bachem et al., 2017;




Clotten et al., 2018). We therefore conclude that this single $\Delta_{47}$-based temperature likely overestimates BWTs at the termination of MIS M2. The otherwise good agreement between the proxies at both study sites adds confidence to our approach and supports the choice of the Mg/Ca calibration and other input parameters for Mg/Ca temperatures.

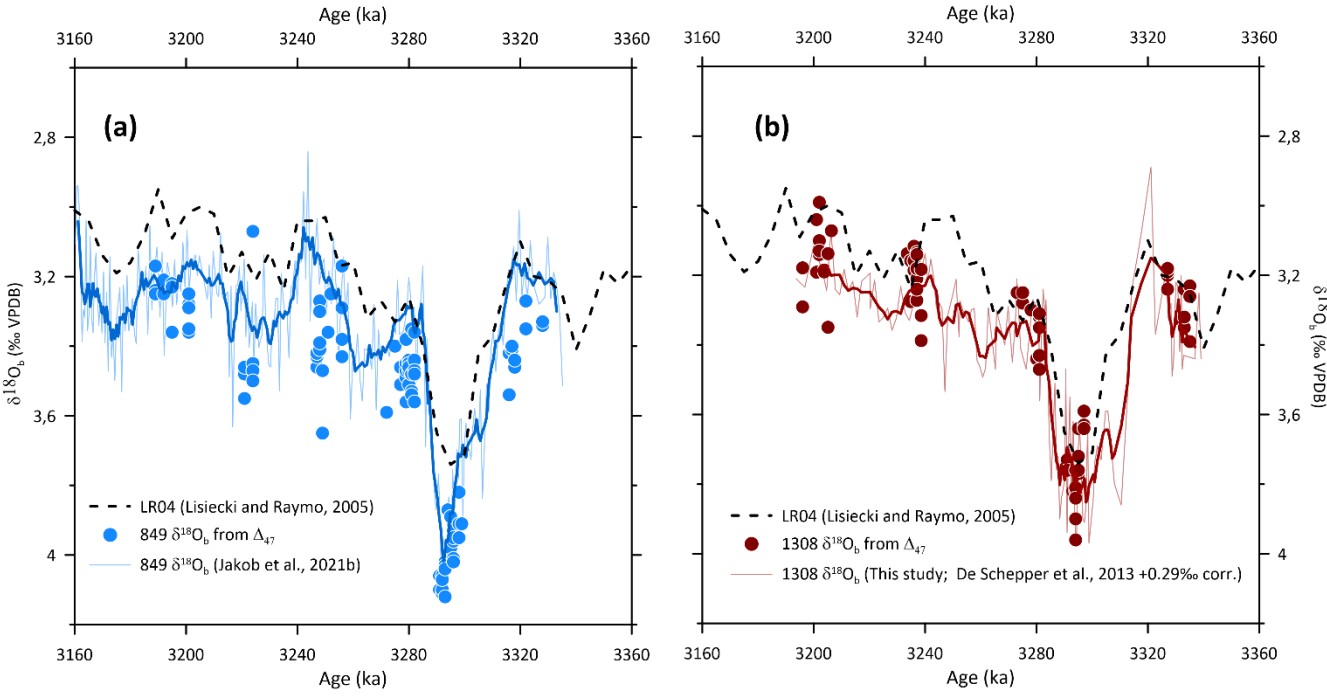

**Figure 4. $\delta^{18}O_b$ from individual $\Delta_{47}$ measurements of Uvigerina spp. and Cibicidoides spp. (normalized to equilibrium following Shackleton et al., 1984)) from (a) Pacific ODP Site 849 and (b) North Atlantic Site U1308**

## 4.2 Warm and salty deep North Atlantic

Previously published North Atlantic BWT reconstructions from the mid-Pliocene have produced contradictory results. While ostracod-based Mg/Ca records from numerous sites in the deep North and South Atlantic suggest mid-Pliocene temperatures were broadly similar to today, at around 2–3°C (Dwyer et al., 1995; Cronin et al., 2005; Dwyer and Chandler, 2009), studies based on Mg/Ca from benthic foraminifera indicate average temperatures of up to ~6°C — more than 3°C warmer than present (Bartoli et al., 2005; Sosdian and Rosenthal, 2009). Our paired benthic foraminiferal Mg/Ca and $\Delta_{47}$ records suggest even warmer temperatures than previously estimated, with both proxies recording average BWTs above 7°C. We note that none of these previously published Mg/Ca-derived BWT records for the Pliocene were adjusted for estimated change in Mg/Ca$_{sw}$. While the record of Sosdian and Rosenthal (2009) (3.150–0 Ma) does not overlap with our new Site U1308 record, recalculating (see Suppl. information and Fig. S6) their data produces average temperatures for their mid–Pliocene interval





closest to our records (3.150–3.0 Ma, ~7.7°C) that are in excellent agreement with our Mg/Ca and $\Delta_{47}$ records. The records of
Bartoli et al. (2005) cannot be recalculated in their published format.

Our new records indicate that BWTs in the deep North Atlantic Ocean were elevated by ~4.5–5.0°C relative to modern,
considerably more than the estimated global average surface warming of 2–3°C relative to the pre-industrial during the MIS
KM5c time slice (McClymont et al., 2020; Haywood et al., 2020). Sea surface temperature reconstructions from the mid-
Pliocene suggest that warming was particularly pronounced in the high northern latitudes, and possibly more so in the North
Atlantic than at comparable latitudes in the Pacific (McClymont et al. 2020). Our records are in fact in good agreement with
the PRISM (Pliocene Research, Interpretation and Synoptic Mapping) dataset, which reconstructs SSTs 4–5°C warmer than
today in the source region of NADW (Dowsett et al., 2009a, b).

**Table 2. Calculated average mid-Pliocene salinity for the North Atlantic (Site U1308) and Pacific (Site 849) using various $\delta^{18}O_{sw}$-S relationships for the modern ocean from [1]LeGrande and Schmidt (2006) and [2]Lynch-Stieglitz et al. (1999).**

| Basin | $\delta^{18}O_{sw}$ (‰) | T (°C) | S min (PSU) | S max (PSU) | Notes |
|-------|-------------------------|--------|-------------|-------------|-------|
| North Atlantic | 1.1 | 7.5 | 36.6 (´North Atlantic[1]´) | 37.0 (´NADW[1]´) | Note that e.g. ´mixed deep water[1]´ and ´all open ocean stations[2]´ give values within this range |
| Pacific | 0.0 | 3.0 | 34.2 (Mixed deep water[1]) | 35.0 (´CDW[1]´) | Note that e.g. ´South Pacific[1]´, ´North Pacific[1]´ and ´all open ocean stations[2]´ give values within this range |

The deep water filling the North Atlantic was not just warmer, but likely also more saline than today. Our average calculated
deep North Atlantic $\delta^{18}O_{sw}$ of 1.1‰ for the mid-Pliocene is considerably higher than its present value of 0.25‰ and similar to
values observed in surface water in the Mediterranean Sea and North Atlantic Subtropical gyre today. Assuming $\delta^{18}O_{sw}$-salinity
relationships established for the modern deep Atlantic (LeGrande and Schmidt, 2006) are valid for the Pliocene, this $\delta^{18}O_{sw}$
value translates to a deep Atlantic salinity of 36.6–37 PSU — up to 2 PSU higher than today (Table 2). Considerably higher
salinity would have aided the formation of warm Northern-sourced deep water by offsetting the buoyancy effect of the elevated
temperature and may help explain how this water mass was sufficiently dense to fill the abyssal Atlantic despite potential
competition from other, colder deep water masses sourced elsewhere. This salinification of the deep North Atlantic is supported
by recent modelling efforts. Weiffenbach et al. (2023) show that an increase in salinity in the high latitude North Atlantic is a
consistent feature in the PlioMIP2 (Pliocene Model Intercomparison Project Phase 2) ensemble. In these models, a closed
Bering Strait and Canadian Archipelago (Haywood et al., 2020) results in dramatically lower in freshwater transport from the
Arctic Ocean into the North Atlantic, which in turn contributes to elevated salinity (~2 PSU) in the Labrador Sea and the





subpolar (40–60°C) North Atlantic. The PlioMIP2 ensemble also simulates lower precipitation-evaporation (P-E) over the
       Labrador Sea and subtropical Atlantic (Feng et al., 2022), further contributing to elevated surface salinity in these regions.

### 4.3 Pacific–Atlantic temperature gradient

The deep Atlantic and Pacific Mg/Ca and $\Delta_{47}$ records confirm a large temperature gradient between these basins during the
mid-Pliocene (Fig. 5a). At the depth of our study sites (~3800 m), bottom water masses bathing the deep North Atlantic today
       are slightly warmer (2.6°C vs. 1.6°C) and saltier (34.9 vs 34.7 PSU) than in the Pacific (Locarnini et al., 2013). The existence
       of a large temperature gradient of up to 4°C between these two basins in the Pliocene was first suggested by Woodard et al.
       (2014). Their foraminiferal Mg/Ca-based BWTs from ODP 1208 suggest temperatures in the North Pacific were, on average,
       ~1.0°C colder (when adjusted for Mg/Ca$_{sw}$ — see suppl. information and Fig. S6) than today during our study interval. At Site
849 we observe warmer BWTs (by ~1.0°C) during the mid-Pliocene compared to today, suggesting that relatively cold deep
       water filled the North Pacific while slightly warmer bottom waters filled the central Pacific. More recently, however, deep-
       water formation in the North Pacific has been suggested for the Pliocene (Burls et al., 2017; Shankle et al., 2021; Ford et al.,
       2022), so it is possible that temperatures at Site 1208 are less representative of the Pacific Ocean as a whole than those at Site
       849.


Nevertheless, our new records substantiate the main interpretation of Woodard et al. (2014) that the deep North Atlantic Ocean
was significantly warmer than the deep Pacific Ocean during the mid-Pliocene (Fig 5a). We also document a large offset in
$\delta^{18}O_{sw}$ between the deep Pacific (0.0‰) and deep North Atlantic Ocean (1.1‰, Fig 5b), likely reflecting a marked salinity
gradient between the saltier Atlantic and fresher Pacific oceans. Testing various $\delta^{18}O_{sw}$-S relationships established for the
modern ocean, we reconstruct salinities of 34.2–35 PSU in the mid-Pliocene deep Pacific (Table 2), suggesting that the Pacific
       Ocean may have been slightly less saline than today. With respect to the ocean-wide salinity budget, a slight freshening of the
       deep Pacific make sense in light of the strong salinification of the much narrower deep North Atlantic inferred from our records.
       We interpret our results to indicate that the deep Pacific and North Atlantic oceans were bathed by water masses with distinctly
       different physical properties during the mid-Pliocene. One possible explanation for the observed temperature gradient is that
mixing was identical to today, but that the Southern Ocean end member cooled enough to compensate for the warm Atlantic
       waters to produce a cold Pacific end result. Given the globally warm surface conditions of the mid-Pliocene, this scenario is,
       however, rather unlikely. Instead, we consider it more likely that limited oceanic exchange occurred between the two basins
       at this time. Although some surface exchange between the Pacific and Atlantic Oceans through the Central American Seaway
       (CAS) may have still occurred in the mid-Pliocene, the seaway was too shallow as this point to significantly influence deep
ocean circulation (e.g. Straume et al., 2020). This suggests a fundamentally different mode of ocean circulation or mixing
       compared





**Figure 5. Comparison of water-mass characteristics at Site U1308 (in red) and Site 849 (in blue). Horizontal stippled lines reflect modern values at the respective locations (a) Reconstructed bottom water temperatures from Mg/Ca and $\Delta_{47}$. Mg/Ca temperatures**





**shown with 5pt and 3pt running means for Site 849 and U1308, respectively. Horizontal error bars on $\Delta_{47}$ temperatures indicate the age range of all individual samples used for each data point. Temperature uncertainties are expressed as solid (68% CI) and stippled (95% CI) bars. (b) calculated $\delta^{18}O_{sw}$ and linear fit (shading = 95% CI ), c) $\delta^{13}C_b$ from 849 (Jakob et al., 2021b) and 1308 (This study; De Schepper et al., 2013).**

to the present. In the modern ocean, heat and salt is distributed from the North Atlantic into the Pacific through the formation

of Circumpolar Deep Water (CDW) in the Southern Ocean. Our results indicate that warm and salty Northern Component Waters (NCW) originating in the North Atlantic was not a major contributor to CDW in the mid-Pliocene — as suggested by Woodard et al. (2014). We suggest that instead of being trapped in the sub-surface, warm NCW upwelled in the Southern Ocean where it possibly contributed to the reduced extent of sea ice (McKay et al., 2012) and the (largely marine based) West Antarctic Ice Sheet (WAIS) in the mid-Pliocene (Naish et al., 2009; Pollard and DeConto, 2009).


Due to the lack of comparable BWT records from the Pacific and Atlantic before 3.3 Ma, it is unclear when the mid-Pliocene circulation state commenced. Determining this would likely provide further clues of the effects it had on the warm climates of the mid-Pliocene. Woodard et al. (2014) suggest that this mode of circulation was abruptly terminated at 2.7 Ma, when the temperature gradient between the North Atlantic and North Pacific was reduced from ~4C° to ~1C°. They therefore suggest

this as a potential contributing factor for iNHG. However, it is unclear if the large gradient between the central Pacific and North Atlantic also ceased at 2.7 Ma. Furthermore, as the North Pacific BWT record ends at 2.5 Ma, it remains unclear if the gradient between this site and the North Atlantic was permanently reduced after iNHG or reemerged at points during the Pleistocene.

## 4.4 Marine Isotope stage M2

Interrupting the warm and mostly stable, high $CO_2$ climate of the mid-Pliocene is Marine Isotope Stage M2 (3.312–3.264 Ma), representing the largest positive benthic oxygen isotope excursion in the Pliocene prior to iNHG (Lisiecki and Raymo, 2005). The ~0.64% increase in the global $\delta^{18}O_b$ stack suggests either a major increase in land ice-volume, strong cooling in the deep ocean, or some combination thereof. Estimates of sea-level fall associated with this event range from relatively minor (Naish and Wilson, 2009; Rohling et al., 2014) to more than 60 meters below modern (Dwyer and Chandler, 2009), consistent with a

very large build-up of ice on land. While the most conservative of these estimates could be explained by an expansion of the Antarctic ice sheet at that time — as is documented in Southern Ocean IRD records (e.g. Passchier, 2011; McKay et al., 2012) — the largest estimate would also require substantial glaciation of the Northern Hemisphere. Although sedimentological evidence suggests at least some ice-growth in the circum-Arctic during M2 (i.e. De Schepper et al., 2014 and references therein), there is no conclusive evidence for the existence of large Northern Hemisphere ice sheets at that time. Deconvolving

the ice-volume signal from the benthic $\delta^{18}O$ record, which would help constrain the extent of glacial expansion and plausible scenarios for ice-sheet growth, has thus far been hampered by a lack of high-resolution BWT reconstructions from this time interval.





Our Pacific Mg/Ca record represents the highest-resolution BWT record for MIS M2 to date and it is the first record of
sufficient temporal resolution to investigate potential leads and lags in the climate system. Our data suggest abrupt MIS M2
cooling of 3–4°C in the deep Central Pacific starting at ~3.30 Ma. This cooling appears to lag changes in benthic $\delta^{18}O$ (Jakob
et al., 2021b, Fig. 2a, measured on the same samples) by approximately 20 kyr, but it is in phase with a decrease in $CO_2$ of
approximately 100 ppm (de la Vega et al., 2020, Fig.2c). We speculate that the early increase in $\delta^{18}O_b$ prior to the onset of M2,
which is lacking a synchronous drop in BWT, represents an increase in global ice volume, and that the subsequent $\delta^{18}O_b$
increase reflects cooling of the deep ocean. Interpreting calculated bottom water $\delta^{18}O_{sw}$ entirely as a measure of ice volume
suggests elevated ice volume (relative to present) during the entire early half of our 849 record (3.334–3.264), i.e., not only
during the MIS M2 cooling event, but also during the preceding interglacial (MIS MG1) (Fig. 2e). Following the termination
of MIS M2, a gradual change in $\delta^{18}O_{sw}$ implies a decrease in ice-volume relative to present. However, the large spatial
differences between North Atlantic and Pacific bottom water $\delta^{18}O_{sw}$ inferred from our records (Fig. 5b) complicates the
interpretation of $\delta^{18}O_{sw}$ as entirely reflective of ice-volume — unless the variations are seen globally.

Our North Atlantic Site U1308 Mg/Ca record is of significantly lower temporal resolution than at Site 849. However, we
observe MIS M2 BWT cooling of a similar amplitude as seen in the Central Pacific. While a lag in cooling relative to $\delta^{18}O_b$
also appears to exist in the North Atlantic (Fig. 3b), this finding is not as robust as in the Pacific. Despite the cooling observed
during MIS M2, North Atlantic BWTs remain warmer than present throughout the event. This is in line with SST
reconstructions from the North Atlantic and Nordic Seas showing that temperatures remained as warm or warmer than the
Holocene average during MIS M2 (Lawrence et al., 2009; Bachem et al., 2017; De Schepper et al., 2013; Naafs et al., 2010).
The sustained high-latitude warmth in the source region of NCW throughout MIS M2 inferred from our records add further
evidence that ice-sheet advance in the Northern Hemisphere was possibly limited to the circum-Arctic.


Based on benthic $\delta^{13}C$ and Nd-isotope records, Kirby et al. (2020) conclude that — unlike during early and late Pleistocene
glacials such as MIS 100 and 2 — NCW largely prevailed in the deep North Atlantic Ocean during M2. This interpretation is
supported by our BWT records showing that the large temperature gradient between the Pacific and North Atlantic was
maintained over M2 (Fig. 4a). Furthermore, fish debris Nd-isotope ratios from North Atlantic Site U1313 (Kirby et al., 2020,
Fig. 3d) suggest that the smaller incursions of Southern Component Waters (SCW) peaked well before $\delta^{18}O_b$ and minima in
BWT and atmospheric $CO_2$, further supporting that an influx of southern waters is not responsible for the observed cooling at
Site U1308. This suggests global-scale cooling during MIS M2, with the source regions for both Southern- and Northern-
sourced deep waters cooling concurrently.

Given the lack of major SCW incursions into the deep North Atlantic, and that cooling in the Pacific appears to be in-phase
with $CO_2$ (de la Vega et al., 2020, Fig. 2c), we interpret our data to indicate that the deep Pacific, and not the Atlantic, was the

ultimate sink for $CO_2$ sequestered from the atmosphere over this event, as was speculated by Kirby et al. (2020). We furthermore suggest the decrease in atmospheric $CO_2$ concentrations as the main driver behind M2 cooling.

At Site 849, our Mg/Ca record reveals additional large changes in deep Pacific temperatures during the mid-Pliocene that have not previously been fully documented. Cooling events of similar amplitude to that of MIS M2 also occur during MIS KM4 and KM6, but these stages are not reflected as significant glacial events in the $\delta^{18}O_b$ record. This suggests that increases in $\delta^{18}O_{sw}$ compensated for the decreases in temperature, leaving $\delta^{18}O_b$ largely unaffected. Cooling during KM4 is associated with a $CO_2$ decrease of similar magnitude to that documented for MIS M2 (~100 ppm). Thus, despite the anomalous $\delta^{18}O_b$ excursion

associated with MIS M2, this event does not appear to not be unique in terms of BWT cooling and atmospheric $CO_2$ decrease in the mid-Pliocene. The amplitude of cooling observed at both sites during MIS M2 suggests that changes in benthic $\delta^{18}O$ associated with this cold stage were mostly driven by temperature change in the deep ocean rather than ice volume.

## 5. Conclusions

We present new benthic foraminiferal Mg/Ca and $\Delta_{47}$ records from Equatorial East Pacific Site 849 and North Atlantic Site

U1308, spanning MIS M2 and the first half of the mid-Pliocene Warm Period. We demonstrate that a large bottom water temperature gradient of >4°C existed between the deep Pacific and Atlantic basins at this time, in line with previous findings, and that the mid-Pliocene deep ocean was less homogenous than at present not just in temperature, but also in $\delta^{18}O_{sw}$ and thus likely salinity. We propose that this was caused by limited oceanic exchange between the deep Atlantic and Pacific oceans, suggesting a fundamentally different mode of ocean circulation and/or mixing than at present. We furthermore find that both

basins cooled by 3–4°C during MIS M2, suggesting that the large positive oxygen isotope excursion associated with this event largely reflects global-scale cooling in the deep ocean—likely driven by decreasing $CO_2$ concentrations — rather than a substantial increase in ice volume on land. Our observation that cooling events of a similar amplitude also occurred in the following two "glacial" Marine Isotope Stages (KM4 and KM6) suggests that that the climatic impact of the M2 event was not as unique as suggested by the benthic $\delta^{18}O$ record alone, and that deep ocean temperatures in the mid-Pliocene were more

variable than has previously been documented.

## Data availability

The data from this paper are archived in the supplement (Table S1-6). In addition, all trace metal data, calculated Mg/Ca and $\Delta_{47}$ temperatures, and new stable isotope data has been submitted to Pangaea (awaiting DOI).

**Author contribution**

AHB, SLH, KAJ, OF and ANM initiated and designed the study. AHB generated and analyzed clumped isotope data under the supervision of EVG and ANM. SLH generated and analyzed clumped isotope data under the supervision of ANM. KAJ generated and analyzed Mg/Ca data under the supervision of OF. All the authors contributed to the palaeoceanographic interpretation. AHB wrote the paper with contributions from all co-authors.

**Competing interests**

The authors declare that they have no conflict of interest.

**Acknowledgements**

This research used samples provided by the International Ocean Discovery Program (IODP) and its predecessor, the Ocean Drilling Program (ODP). Jordan Donn Holl and Lubna Al-Saadi (University of Bergen), Phillipp Geppert, Lena Heiler, Franz Kerschofer and Verena Schreiber (Heidelberg University) helped with sample processing. Enver Alagoz (University of 525 Bergen), Silvia Rheinberger and Christian Scholz (Heidelberg University) are thanked for laboratory assistance during clumped isotope and Mg/Ca analysis, respectively. Irene Heggstad (University of Bergen) is acknowledged for SEM assistance. We thank Alvaro Fernandez Bremer for sharing Matlab code for error propagation. The authors acknowledge the financial support from the European Research Council (ERC), the German Research Foundation (DFG) and the Trond Mohn Foundation.

**Financial support**

Funding for this work was provided by the European Research council (grant no. 638467 to ANM), the Trond Mohn Foundation (grant no. BFS2015REK01 to ANM) and the German Research Foundation (grants JA2803/2–1 to KAJ, FR2544/6 to OF)




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
