# Peer review of "Limited exchange between the deep Pacific and Atlantic oceans during the warm mid-Pliocene and MIS M2 "glaciation""

_Climate of the Past, 2023_

## Author Comment (AC1)

Hauge Braaten et al. present a combination of thermometers (Mg/Ca and D47) applied to benthic foraminifera from 2 sites (in the Pacific Ocean and in the Atlantic Ocean) to reconstruct the bottom water interaction between the 2 oceans during the mid Piacenzian. This article presents a significant amount of measurements and high quality conclusions. It deserves to be published. I have a few suggestions, mostly on salinity reconstructions and minor comments.

Regarding the salinity reconstruction, I would suggest being more critical of your methodology. The relationship between d18Osw and salinity changes over time. Assuming that the relationship during the mid Piacenzian is similar to today's is probably wrong. Although the authors mentioned it, they assume that their reconstructions of salinity is correct and I suggest being more careful in interpreting your data. Additionally, the authors should add a discussion of the uncertainties in thier estimated salinity. Is a change of 2 PSU, using this methodology, really significant? (Line 377).

We recognize the inherent uncertainties in calculating salinity from d18O for the Pliocene given the likely changes in d18O-salinity relationships over time as well as uncertain mean ocean d18O. Our calculations were therefore meant more as a sanity check rather than reliable estimates of salinity. Given the reviewers comments, however, we suggest reducing the perceived emphasis on absolute values for salinity by omitting these calculations altogether, and simply stating that our d18Osw data suggests a large salinity gradient between the two basins.

Same question for a salinity difference of 0.2 between the Atlantic Ocean and the Pacific Ocean (Line 391).

This value refers to the observed salinity gradient between Pacific and Atlantic oceans today.

Finally, did the authors take into account the global changes during this period (based on sea level change for example) in order to obtain the local salinity signal?

See our reply above. This part of the discussion will be shortened and clarified.

The preservation and contamination tests presented in supplementary material are not mentioned in the text. I found it confusing to only read that the foraminifera are well preserved without any justification, until I read the supplementary material. I would suggest, at least, mentioning the supplementary material.

We had included references to the supplementary information about the contamination tests (line 193-194) and SEM images (line 144 and 215), but we will adjust the main text to more explicitly state where information about the preservation state can be found.

Lines 209, 254 and 255: do the authors have uncertainties at 1 or 2 sigma?

We will update the text to specify that the values refer to uncertainties at the 2SD level and will add propagated uncertainties on the Mg/Ca temperatures to Figs. 2, 3 and 5.

Line 281: I would mention that the author are still talking about Mg/Ca derived temperature in this paragraph.

We will clarify that this line refers to the Mg/Ca-derived temperature record.

Line 295: MIS not MI2

Thank you for noticing this error. We will fix this.

Figure 2 and 3: It would be great if the authors could also add the uncertainties on the Mg/Ca temperatures

Please see our response above.

Lines 395-396: Can the authors expand on this sentence, please?

The previously published Mg/Ca record from Site 1208 (~3350 m) in the North Pacific produces temperatures that are, on average, somewhat colder than present whereas our Central Pacific record from the deeper Site 849 (~3850 m) suggests temperatures that are somewhat warmer than present. We therefore speculated that Site 1208 may be more directly influenced by deep waters forming in the North Pacific (as is hypothesized for the Pliocene), driving the observed temperature difference between these two Pacific records. However, this would imply that the North Pacific sourced water was colder than the (likely southern) source bathing Site 849. Furthermore, the absolute temperatures derived from Mg/Ca at Site 1208 depend strongly on the selected calibration. With only the Mg/Ca record available for that site, there is currently no independent way to cross-check the results like we were able to do at Site 849. We can also not easily transfer the calibration we used at Site 849 to Site 1208 since that record is based on a different species of benthic foraminifera (*Uvigerina spp.*). As a result, we will clarify this part but would also like remain more vague and rephrase this section to: "The existence of a large temperature gradient up to 4°C between these two basins in the Pliocene was first suggested by Woodard et al. (2014). Their foraminiferal Mg/Ca based BWTs from ODP Site 1208 suggests temperatures in the North Pacific were, on average, ~1°C colder (when adjusted for Mg/Ca$_{sw}$ — see suppl. information and Fig. S6) than today during our study interval. In contrast, at Site 849 we observe warmer BWTs (by 1°C) during the mid-Piacenzian compared to today. At face value this discrepancy would suggest that different water masses influenced the North Pacific and the central Pacific. As deep water formation in the North Pacific has been suggested for the Pliocene (Burls et al., 2017; Shankle et al., 2020; Ford et al., 2022), it is possible that temperatures at Site 1208 were more directly influenced by this water mass and are thus less representative of the Pacific Ocean as a whole than those at Site 849. It is however important to keep in mind that the absolute Mg/Ca-based temperatures are highly calibration dependent and are currently only cross-validated with $\Delta_{47}$ at our study sites."

Line 410: What are the authors referring to? Model, data? References are missing

This statement does not refer to a specific study but is a speculative explanation, based on end-member mixing, for how we could in principle end up with a large temperature gradient between the Pacific and North Atlantic while maintaining a modern circulation

state. We suggest to make this more clear by rephrasing to "Hypothetically, a possible explanation for the observed temperature gradient, if water mass mixing was identical to today, could be that the Southern Ocean end member cooled enough to compensate for the warm Atlantic waters to produce a cold Pacific end result. Given the globally warm surface conditions of the mid-Piacenzian, this scenario is, however, rather unlikely."

Lines 458-462: How did the authors conclude on the ice volume proportion of thier d18Osw parameters? How did they remove the salinity part of the d18Osw, as explaining previously?

Please see our reply above; we suggest omitting the salinity calculations to avoid putting unintended emphasis of these very uncertain estimates.

---

## Author Comment (AC3)

The authors of this study present Mg\Ca and clumped isotope bottom water temperature records for the Late Pliocene from Site 849 (3346m) in the Pacific, and U1308 (3426m) in the Atlantic. They find that the Atlantic site was warmer (>4C as compared to ~1C warmer in the modern Ocean) and saltier than the Pacific site during this time. The authors attribute this to a different ocean circulation regime under which there is a more limited water mass exchange between the deep Atlantic and Pacific basins. Based on the amplitude of the cooling observed at both sites during MIS M2 they suggest that the benthic delta18O changes associated with this cold stage were mostly driven by changes in the deep ocean temperature rather than ice volume.

Overall, I found the new records exciting and very insightful and the manuscript rigorous and well structured. As a modeler I am not in the position to provide a critical assessment of the analysis methods but can offer an assessment of the dynamical interpretation. While the authors are careful to be vague as to mechanism in the abstract and conclusions, I am not completely convinced of the tentative explanation for the larger difference in water mass properties between the two basins stated in the discussion section (Ln 427-429). Given the circumpolar nature of the Southern Ocean and that already ~80% of modern deep water is upwelled in the Southern Ocean it is not clear how this would be enhanced in the Pliocene and lead to a reduction in the inter basin exchange of deep water. It would be good if the authors could point to a study with ocean simulations that reproduce the proposed type of circulation regime change. On the other hand, Pliocene simulations with north Pacific deep water formation result in fresher North Pacific deep waters relative to preindustrial (Burls et al., 2017, Fig. 5) and reduced warming relative to the Atlantic (Burls et al., 2017, Fig. 4). So it seems both scenarios should be considered in the discussion.

We will note that the influence of North Pacific Deep Water on Site 849 offers another possible explanation for the large temperature gradient observed between the Pacific and Atlantic basins. Given the reviewer's comment on the likeliness of increased upwelling of the warmer NADW in the Southern Ocean, we suggest omitting this speculative part of the section. We suggest revising this paragraph as follows: "Hypothetically, a possible explanation for the observed temperature gradient, if water mass mixing was identical to today, could be that the Southern Ocean end member cooled enough to compensate for the warm Atlantic waters to produce a cold Pacific end result. Given the globally warm surface conditions of the mid-Piacenzian, this scenario is, however, rather unlikely. Another possibility is that the deep central Pacific was bathed by water masses sourced from the North Pacific, rather than from the Southern Ocean. While formation of NPDW has been suggested for the Pliocene (Burls et al., 2017; Shankle et al., 2021; Ford et al., 2022), the modelled spatial extent of NPDW during the mPWP does not support a large influence of this water mass on the abyssal central Pacific (Ford et al., 2022). Instead, we consider it most likely that limited oceanic exchange occurred between the Pacific and Atlantic basins at this time. This suggests a fundamentally different mode of ocean circulation or mixing compared to the present."

The backing out of salinity estimates is a nice part of the manuscript but as one of the other reviewers mentions some more details are needed explaining how the ice sheet contribution was handled. This should not affect the basin gradient though. The limitations/robustness of assuming modern relationships should be discussed e.g. see Fig. S7 in Gaskell et al., (2022).

We recognize the inherent uncertainties in calculating salinity from d18O for the Pliocene given the likely changes in d18O-salinity relationships over time. To avoid putting too much

emphasis on absolute values, we suggest omitting these calculations, and simply stating that there is likely to be a large salinity gradient between the two basins given the reconstructed difference in d18Osw.

Minor comments:

Ln 405-406:"Slightly less saline" perhaps add modern values to Table 2 so that the reader can assess just how much fresher for the Pacific.

See above, we will remove Table 2.

Fig 2d: The incorrect axis label is provided; it should be insolation and presumably units of W/m^2.

We will fix the axis label on Fig. 2d.

Fig 2e: Why isn't the 5pt running mean shown as in the other panels?

We will add the 5pt running mean to Fig. 2e.

Fig 5c: Modern d13C values are missing and would be helpful for reference.

We will add modern $\delta^{13}$C values to Fig. 5c.

---

## Author Response (AR1)

**Reviewer 1**

Hauge Braaten et al. present a combination of thermometers (Mg/Ca and D47) applied to benthic foraminifera from 2 sites (in the Pacific Ocean and in the Atlantic Ocean) to reconstruct the bottom water interaction between the 2 oceans during the mid Piacenzian. This article presents a significant amount of measurements and high quality conclusions. It deserves to be published. I have a few suggestions, mostly on salinity reconstructions and minor comments.

Regarding the salinity reconstruction, I would suggest being more critical of your methodology. The relationship between d18Osw and salinity changes over time. Assuming that the relationship during the mid Piacenzian is similar to today's is probably wrong. Although the authors mentioned it, they assume that their reconstructions of salinity is correct and I suggest being more careful in interpreting your data. Additionally, the authors should add a discussion of the uncertainties in thier estimated salinity. Is a change of 2 PSU, using this methodology, really significant? (Line 377).

We recognize the inherent uncertainties in calculating salinity from d18O for the Pliocene given the likely changes in d18O-salinity relationships over time as well as uncertain mean ocean d18O. Our calculations were therefore meant more as a sanity check rather than reliable estimates of salinity. Given the reviewers comments, however, we reduce the perceived emphasis on absolute values for salinity by omitting these calculations altogether, and simply state that our d18Osw data suggests a large salinity gradient between the two basins (Lines 387-389).

Same question for a salinity difference of 0.2 between the Atlantic Ocean and the Pacific Ocean (Line 391).

This value refers to the observed salinity gradient between Pacific and Atlantic oceans today. We have not made changes to this sentence

Finally, did the authors take into account the global changes during this period (based on sea level change for example) in order to obtain the local salinity signal?

See our reply above. This part of the discussion has been shortened and clarified

The preservation and contamination tests presented in supplementary material are not mentioned in the text. I found it confusing to only read that the foraminifera are well preserved without any justification, until I read the supplementary material. I would suggest, at least, mentioning the supplementary material.

We had included references to the supplementary information about the contamination tests (line 193-194) and SEM images (line 144 and 215), but we have adjusted the main text to more explicitly state where information about the preservation state can be found (lines 135, 147 and 222 in revised manuscript).

Lines 209, 254 and 255: do the authors have uncertainties at 1 or 2 sigma?

We have updated the text throughout to specify that the values refer to uncertainties at the 2σ level and have added propagated uncertainties on the Mg/Ca temperatures to Figs. 2, 3 and 5.

Line 281: I would mention that the author are still talking about Mg/Ca derived temperature in this paragraph.

We have clarified that this line refers to the Mg/Ca-derived temperature record (line 280 in revised manuscript)

Line 295: MIS not MI2

We have fixed this

Figure 2 and 3: It would be great if the authors could also add the uncertainties on the Mg/Ca temperatures

Please see our response above.

Lines 395-396: Can the authors expand on this sentence, please?

The previously published Mg/Ca record from Site 1208 (~3350 m) in the North Pacific produces temperatures that are, on average, somewhat colder than present whereas our Central Pacific record from the deeper Site 849 (~3850 m) suggests temperatures that are somewhat warmer than present. We therefore speculated that Site 1208 may be more directly influenced by deep waters forming in the North Pacific (as is hypothesized for the Pliocene), driving the observed temperature difference between these two Pacific records. However, this would imply that the North Pacific sourced water was colder than the (likely southern) source bathing Site 849. Furthermore, the absolute temperatures derived from Mg/Ca at Site 1208 depend strongly on the selected calibration. With only the Mg/Ca record available for that site, there is currently no independent way to cross-check the results like we were able to do at Site 849. We can also not easily transfer the calibration we used at Site 849 to Site 1208 since that record is based on a different species of benthic foraminifera (*Uvigerina spp.*). As a result, we have clarifed this part and have rephrased this section to: "The existence of a large temperature gradient of up to 4°C between these two basins in the Pliocene was first suggested by Woodard et al. (2014). Their foraminiferal Mg/Ca-based BWTs from ODP 1208 suggest temperatures in the North Pacific were possibly somewhat colder than today (when adjusted for Mg/Ca$_{sw}$ — see suppl. information and Fig. S6) during the mid-Piacenzian. In contrast, at Site 849 we observe warmer BWTs during the mid-Piacenzian compared to today. At face value, this discrepancy would suggest that different water masses influenced the North Pacific and the central Pacific. As deep-water formation in the North Pacific has been suggested for the Pliocene (Burls et al., 2017; Shankle et al., 2021; Ford et al., 2022), it is possible that temperatures at Site 1208 were more directly influenced by this water mass and are thus more representative of the Pacific Ocean as a whole than those at Site 849. It is however important to keep in mind that the absolute Mg/Ca-based

temperatures are highly calibration dependent and are currently only cross-validated with $\Delta_{47}$ at our study sites..”

Line 410: What are the authors referring to? Model, data? References are missing

This statement does not refer to a specific study but is a speculative explanation, based on end-member mixing, for how we could in principle end up with a large temperature gradient between the Pacific and North Atlantic while maintaining a modern circulation state. We have clarified this by rephrasing to “Hypothetically, a possible explanation for the observed temperature gradient, if water mass mixing was identical to today, could be that the Southern Ocean end member cooled enough to compensate for the warm Atlantic waters to produce a cold Pacific end result. Given the globally warm surface conditions of the mid-Piacenzian, this scenario is, however, rather unlikely.”

Lines 458-462: How did the authors conclude on the ice volume proportion of thier d18Osw parameters? How did they remove the salinity part of the d18Osw, as explaining previously?

Please see our reply above; we have omitted salinity calculations to avoid putting unintended emphasis of these very uncertain estimates.

**Reviewer 2**

In this manuscript, Braaten et al. present a multi-proxy record of deep ocean temperature and d18O of seawater over the mid-Pliocene warm period and the M2 glaciation in the North Atlantic and Equatorial Pacific. I found the manuscript well written, and they presented compelling evidence that the Atlantic and the Pacific were quite different in temperature and d18O of seawater. The implications of their records are that the characteristics of deep-water formation were very different during the Pliocene with the northern component water being very warm and very salty.

I have two moderate requests for the authors.

In the supplemental, the authors present cross plots of the El/Ca to investigate possible contamination. Could you also please include the El/Ca and Mg/Ca with time on the x-axis? Are there temporal trends in the Mg/Ca data that could be explained by contamination? The reconstructed temperatures here and Jakob et al., 2020 are quite high so it would be helpful to know if there are temporal patterns.

Please see figure below (upper panels are from Site 849, lower panels from U1308) with Mg/Ca, Fe/Ca and Mn/Ca vs. time. We do not observe any obvious temporal trends in the Mg/Ca data that can be explained by the minor element ratios. At Site 849, there is a slight downcore increasing trend in both the Fe/Ca and Mn/Ca values, but this is less evident in the Mg/Ca values.

[Figure]

The authors did a nice job of propagating the error on the clumped isotope measurements, but the Mg/Ca did not have the same consideration. Could you please estimate the error on the temperature and d18Osw calculations? I usually use PSUSolver by Thirumalai et al 2016 (doi:10.1002/2016PA002970) which you can find here: https://www.mathworks.com/matlabcentral/fileexchange/59565-paleo-seawater-uncertainty-solver

We thank the reviewer for suggestion and have added the propagated errors on the calculated Mg/Ca temperatures and $\delta^{18}O_{sw}$ to Figs. 2, 3 and 5.

Minor comment:

Figure 2, (d) panel labeled for eNd

Thank you for spotting this, we have fixed the axis label on Fig. 2d.

**Reviewer 3**

The authors of this study present Mg\Ca and clumped isotope bottom water temperature records for the Late Pliocene from Site 849 (3346m) in the Pacific, and U1308 (3426m) in the Atlantic. They find that the Atlantic site was warmer (>4C as compared to ~1C warmer in the modern Ocean) and saltier than the Pacific site during this time. The authors attribute this to a different ocean circulation regime under which there is a more limited water mass exchange between the deep Atlantic and Pacific basins. Based on the amplitude of the cooling observed at both sites during MIS M2 they suggest that the benthic delta18O changes

associated with this cold stage were mostly driven by changes in the deep ocean temperature rather than ice volume.

Overall, I found the new records exciting and very insightful and the manuscript rigorous and well structured. As a modeler I am not in the position to provide a critical assessment of the analysis methods but can offer an assessment of the dynamical interpretation. While the authors are careful to be vague as to mechanism in the abstract and conclusions, I am not completely convinced of the tentative explanation for the larger difference in water mass properties between the two basins stated in the discussion section (Ln 427-429). Given the circumpolar nature of the Southern Ocean and that already ~80% of modern deep water is upwelled in the Southern Ocean it is not clear how this would be enhanced in the Pliocene and lead to a reduction in the inter basin exchange of deep water. It would be good if the authors could point to a study with ocean simulations that reproduce the proposed type of circulation regime change. On the other hand, Pliocene simulations with north Pacific deep water formation result in fresher North Pacific deep waters relative to preindustrial (Burls et al., 2017, Fig. 5) and reduced warming relative to the Atlantic (Burls et al., 2017, Fig. 4). So it seems both scenarios should be considered in the discussion.

We have noted in the text that the influence of North Pacific Deep Water on Site 849 offers another possible explanation for the large temperature gradient observed between the Pacific and Atlantic basins. Given the reviewer's comment on the likeliness of increased upwelling of the warmer NADW in the Southern Ocean, we have omitted this speculative part of the section. We have revised this paragraph as follows: "Hypothetically, one possible explanation for the observed temperature gradient, if water mass mixing was identical to today, could be that the Southern Ocean end member cooled enough to compensate for the warm Atlantic waters to produce a cold Pacific end result. Given the globally warm surface conditions of the mid-Pliocene, this scenario is, however, rather unlikely. Another possibility is that the deep central Pacific was bathed by a water mass sourced from the North Pacific rather than from the Southern Ocean. While formation of NPDW has been suggested for the Pliocene (Burls et al., 2017; Shankle et al., 2021; Ford et al., 2022), the modelled spatial extent of NPDW during the mPWP does not support a large influence of this water mass on the abyssal central Pacific (Ford et al., 2022). Instead, we consider it more likely that limited oceanic exchange occurred between the two basins at this time. This suggests a fundamentally different mode of ocean circulation or mixing compared to the present, where heat and salt is distributed from the North Atlantic into the Pacific"

The backing out of salinity estimates is a nice part of the manuscript but as one of the other reviewers mentions some more details are needed explaining how the ice sheet contribution was handled. This should not affect the basin gradient though. The limitations/robustness of assuming modern relationships should be discussed e.g. see Fig. S7 in Gaskell et al., (2022).

We recognize the inherent uncertainties in calculating salinity from d18O for the Pliocene given the likely changes in d18O-salinity relationships over time. To avoid putting too much emphasis on absolute values, we have omitted these calculations, and simply state that there is likely to be a large salinity gradient between the two basins given the reconstructed difference in d18Osw (Lines 387-389).

Minor comments:

Ln 405-406:"Slightly less saline" perhaps add modern values to Table 2 so that the reader can assess just how much fresher for the Pacific.

See above, we have removed Table 2.

Fig 2d: The incorrect axis label is provided; it should be insolation and presumably units of W/m^2.

We have fixed the axis label on Fig. 2d.

Fig 2e: Why isn't the 5pt running mean shown as in the other panels?

We have added the 5pt running mean to Fig. 2e.

Fig 5c: Modern d13C values are missing and would be helpful for reference.

We have added modern $\delta^{13}C$ values to Fig. 5c.